

# Artificial liver research output and citations from 2004 to 2017: a bibliometric analysis

Yan Li[1,*], Meizhi He[1,*], Ziyuan Zou[2], Xiaohui Bian[1], Xiaowen Huang[2], Chen Yang[2], Shuyi Wei[1] and Shixue Dai[3,4]

[1] The Second School of Clinical Medicine, Southern Medical University, Guangzhou, Guangdong, China
[2] The First School of Clinical Medicine, Southern Medical University, Guangzhou, Guangdong, China
[3] Department of Gastroenterology, Guangdong General Hospital and Guangdong Academy of Medical Sciences, South China University of Technology, Guangzhou, Guangdong, China
[4] Guangdong Geriatrics Institute, Guangdong General Hospital and Guangdong Academy of Medical Sciences, South China University of Technology, Guangzhou, Guangdong, China
* These authors contributed equally to this work.

Corresponding author
Shixue Dai, shixuedai@hotmail.com

## ABSTRACT

**Background:** Researches on artificial livers greatly contribute to the clinical treatments for liver failure. This study aimed to evaluate the research output of artificial livers and citations from 2004 to 2017 through a bibliometric analysis.

**Methods:** A list of included articles on artificial livers were generated after a comprehensive search of the Web of Science Core Collection (from 2004 to 2017) with the following basic information: number of publications, citations, publication year, country of origin, authors and authorship, funding source, journals, institutions, keywords, and research area.

**Results:** A total of 968 included articles ranged from 47 citations to 394 citations with a fluctuation. The publications were distributed in 12 countries, led by China ($n = 212$) and the US ($n = 207$). There were strong correlations of the number of citations with authors ($r^2 = 0.133$, $p < 0.001$), and countries ($r^2 = 0.275$, $p < 0.001$), while no correlations of the number of citations with the years since publication ($r^2 = 0.016$, $p = 0.216$), and funding ($r^2 < 0.001$, $p = 0.770$) were identified. Keyword analysis demonstrated that with the specific change of "acute liver failure," decrease in "bioartificial livers" and "hepatocyte," and increase in "tissue engineering" were identified. The top 53 cited keyword and keyword plus (including some duplicates counts) were identified, led by bioartificial liver (405 citations) and hepatocyte (248 citations). The top 50 cited keywords bursts were mainly "Blood" (2004–2008), "hepatocyte like cell" (2008–2015), and "tissue engineering" (2014–2017). All keywords could be classified into four categories: bioartificial livers (57.40%), blood purification (25.00%), clinical (14.81%), and other artificial organs (2.78%).

**Discussion:** This study shows the process and tendency of artificial liver research with a comprehensive analysis on artificial livers. However, although it seems that the future of artificial livers seems brighter for hepatocyte transplantation, the systems of artificial livers now are inclined on focusing on blood purification, plasma exchange, etc.

# INTRODUCTION

Owing to the high mortality of liver failure, numerous studies had investigated its treatment. Artificial livers were a promising kind of treatment for liver failure. In clinical practice, different kinds of bioartificial systems, blood purification and hepatocyte transplantation had already been applied in liver failure (*Naruse, Tang & Makuuch, 2007*; *Wallin, 2005*). However, a great number of obstacles and setbacks were experienced in investigating artificial livers (*Carpentier, Gautier & Legallais, 2009*).

In 1992 and 1994, *Ash et al. (1992)* and *Ash (1994)* set a precedent in performing the first liver dialysis which was previously named as Biologic-DT. Subsequently, Stange and Mitzner (*Stange et al., 2000*) initially developed the Molecular Adsorbent Recirculating System (MARS). Furthermore, various bioartificial liver systems had been reported in nine clinical studies since 1990 (*Park & Lee, 2005*). Although the artificial liver was considered as one of the most effective treatment for liver failure, the definition of an artificial liver still remains obscure.

Additionally, in 2006, *Onodera et al. (2006)* classified artificial livers into three categories: bioartificial liver, blood purification, and hepatocyte transplantation. In the early works, liver support devices were considered as the treatments of liver failure, and these devices have developed into two different strategies: blood purification and bioartificial liver. The purpose of the former was the removal of toxins in the blood related to coma and cerebral edema, while the latter's intention was to provide metabolic, detoxic, and synthetic function of hepatocytes. However, both of these devices were restricted to the lack of both suitable animal models of liver failure and complete understanding of the pathophysiology of liver failure. Thus, hepatocyte transplantation had developed.

Bibliometric sciences provide a statistical and quantitative analysis of publications and offer a convenient way to visibly measure researchers' efforts in the investigation of a specific field (*Ashok et al., 2016*; *Yao et al., 2018*). Though bibliometric methods aim to make comments on qualitative features, the major purpose of their analysis is to transform something intangible (scientific quality) into a manageable entity (*Wallin, 2005*). Citations analysis is a bibliometric process that determine the influence of an article and make further explanations according to the original information about authors and journals (*Schmidt et al., 2014*). The more cited an article is, the greater influence it makes in a specific area (*Eyre-walker & Stoletzki, 2013*). Despite the inevitable limitations (*Wallin, 2005*) of bibliometric analysis in assessing research quality, bibliometric analysis is widely known as one of the best measurements of research trend (*Zhou et al., 2018*).

This study is the first multicenter retrospective study on artificial livers with bibliometric analysis, based on *Onodera et al.'s (2006)* definition of artificial livers. It aimed at evaluating artificial livers output and citations from 2004 to 2017 with a bibliometric analysis, which helps researchers understand the development process of artificial livers and provide guidance for the future direction of further researches.
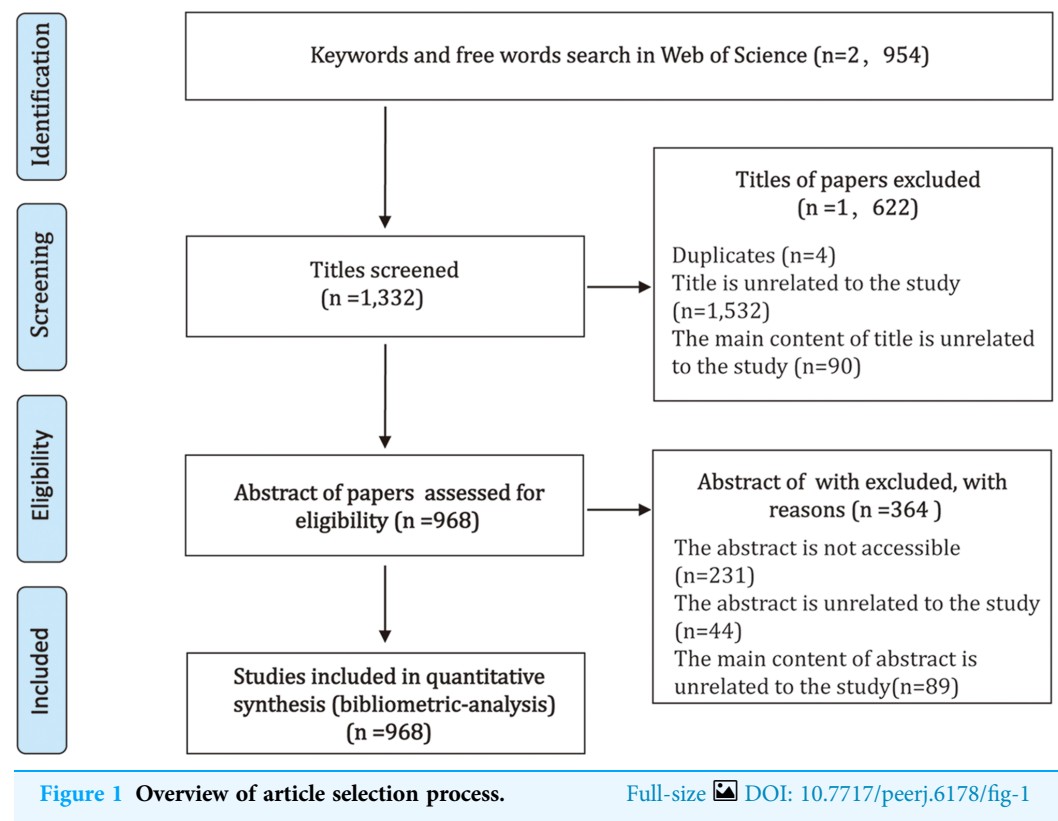

**Figure 1 Overview of article selection process.**

# MATERIALS AND METHODS

## Search strategy

With strict inclusion criteria, Web of Science (WoS) provides adequate researches for literature analysis, including keywords, authors, institutions, countries, and publication years, which were vital for bibliometric analysis. Therefore, we conducted a citation search of the WoS Core Collection from 2004 to 2017. The keywords and free words were "Bioartificial Liver" or "Bioartificial Livers" or "Liver, Bioartificial" or "Livers, Bioartificial" or "Artificial Liver" or "Artificial Livers" or "Livers, Artificial" on topic field. All retrieval was conducted on 25th May 2018 to avoid possible changes in citation rate. After all data were extracted, articles were ranked by citation number.

## Inclusion and exclusion

Two independent investigators (M.Z. He and X.H. Bian), respectively, screened all the titles and abstracts to select eligible articles according to the inclusion criteria (Fig. 1). Any questions were resolved by discussion and the help of the third independent investigator (Y. Li). Data were retrieved from literature if meeting the following criteria: (1) the main content should be related to the study; (2) literature could be any type of researches (such as article, review, editorial material, and meeting abstract); (3) literature involving other topics related to the study. The exclusion criteria were: (1) duplicates; (2) the titles and abstract of papers were not related to the study; (3) the abstract of paper was not accessible.

## Evaluation of included articles

The following data were extracted from the 968 included articles by one investigator (M.Z. He) as followed: (1) the number of publications; (2) citations; (3) publication year; (4) country of origin; (5) authors and authorship; (6) funding source, (7) journals; (8) institutions; (9) keywords and research fields. Articles that were collaborative work of authors from multiple countries were also identified. In addition, the top 18 cited (T18) journals of the top 100 cited (T100) articles were evaluated with impact factors (Ifs) which were in accordance with the 2017 edition of journal citation reports: science edition (2017–2018).

## Statistical analysis

With GraphPad Prism version 6.0, the Spearman test was used to evaluate the strength and direction of the linear relationship between the number of T100 citations and the number of authors, year since publication, funding, and countries in each publication. Furthermore, the Spearman test with GraphPad Prism was also applied to evaluate the correlation of citation index between different databases (WoS Core Collection and Scopus) and the correlations of the number of impact factor with the number of T100 articles per journals and the number of average citations. The citation index was measured as the true impact of an article independent of short-lived trends (*Liu et al., 2016*) and the impact factor usually serves as indicator for reflecting the average number of yearly citations for recent papers published in the journal (*Garfield, 2006*). Both of them were used to assess the equality of research output in most bibliometric analyses. All probability values were two-tailed, meanwhile, the threshold of the number of T100 citations, countries, and authors for significance was set at $p < 0.001$, while others were not.

## Bibliometric analysis

Using the online analysis platform of literature metrology (http://bibliometric.com/), the trend of the number of publications, the trend of country of origin, the cooperation between countries, and the trend of keywords were shown. All the data were standardized by the frequency of occurrence.

With CiteSpace, the network map of authors, co-authors and institutions were shown by publication year, and the number of citations via citation-tree rings, while the network map of keywords and keyword plus were shown only according to number of citations (*Chen, 2004*).

## RESULTS

### Total number of published items

The number of published items on artificial livers was considered as an index of research productivity. A total of 2,954 papers were identified after the initial record in the WoS from 1986 to 2017. From 2,954 records, 1,622 titles were excluded, because 1,532 titles were unrelated to artificial livers, 90 titles were not mainly discussing artificial livers, and four titles were duplicates. In addition, 364 articles were excluded, because 231 abstracts of which were not accessible, 89 abstracts were not mainly discussing artificial

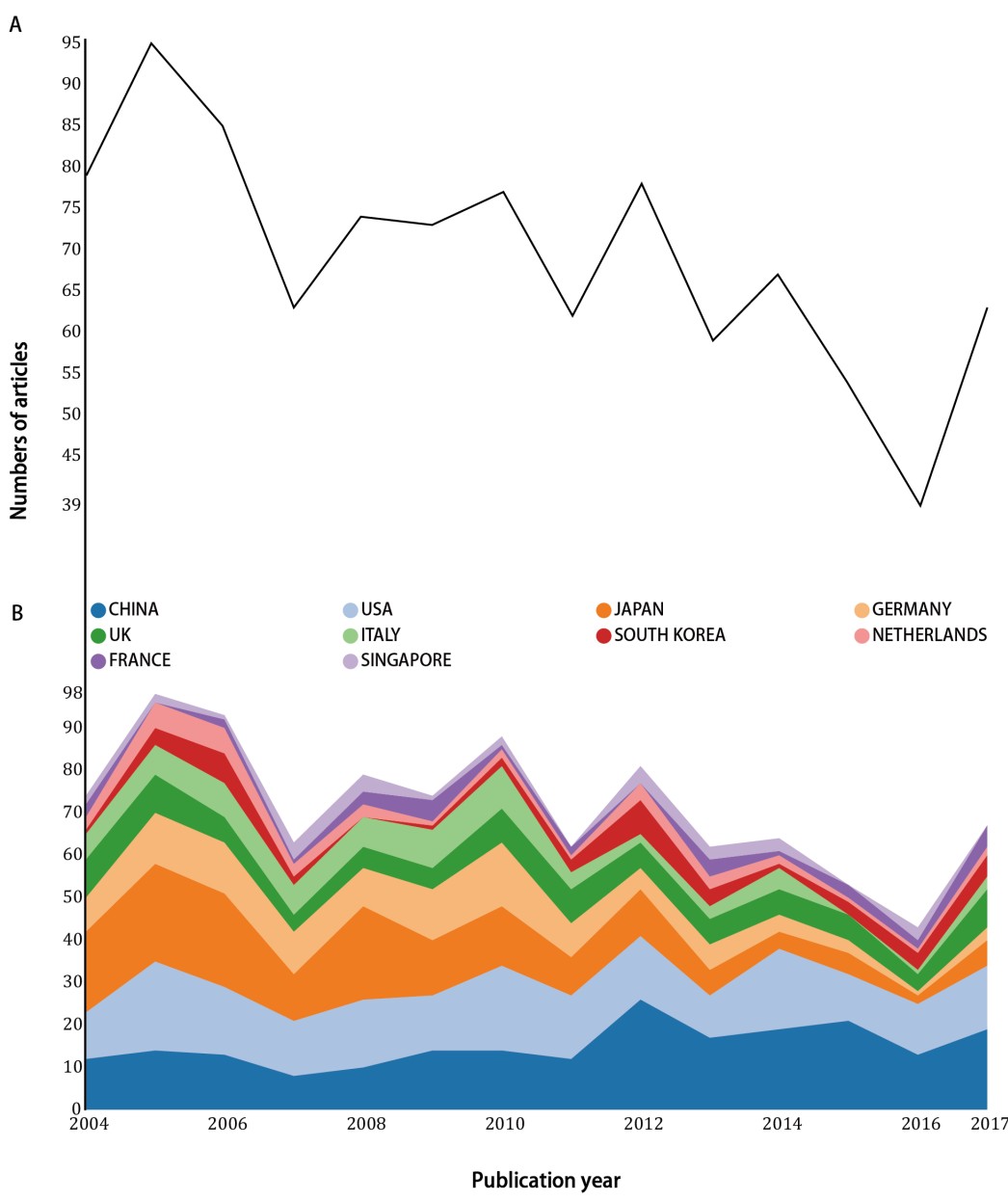

**Figure 2 Numbers of included articles (A) and the growth trends of countries (B) from 2004 to 2017.**

livers, and 44 abstracts were unrelated to artificial livers. Thus, 968 articles were extracted in WoS during the period from 2004 to 2017. The number of articles reached a peak of 95 in 2005 and rapidly decreased to a nadir of 63 in 2007, then fluctuated up and down during 2008–2015 (Fig. 2A). However, it reached the lowest point in 2016 and grew again in 2017.

## Country of origin

A total of 12 different countries contributed to the literature on artificial livers (Fig. 2B). China and the US produced similar number of articles during 2004–2017, while a decrease

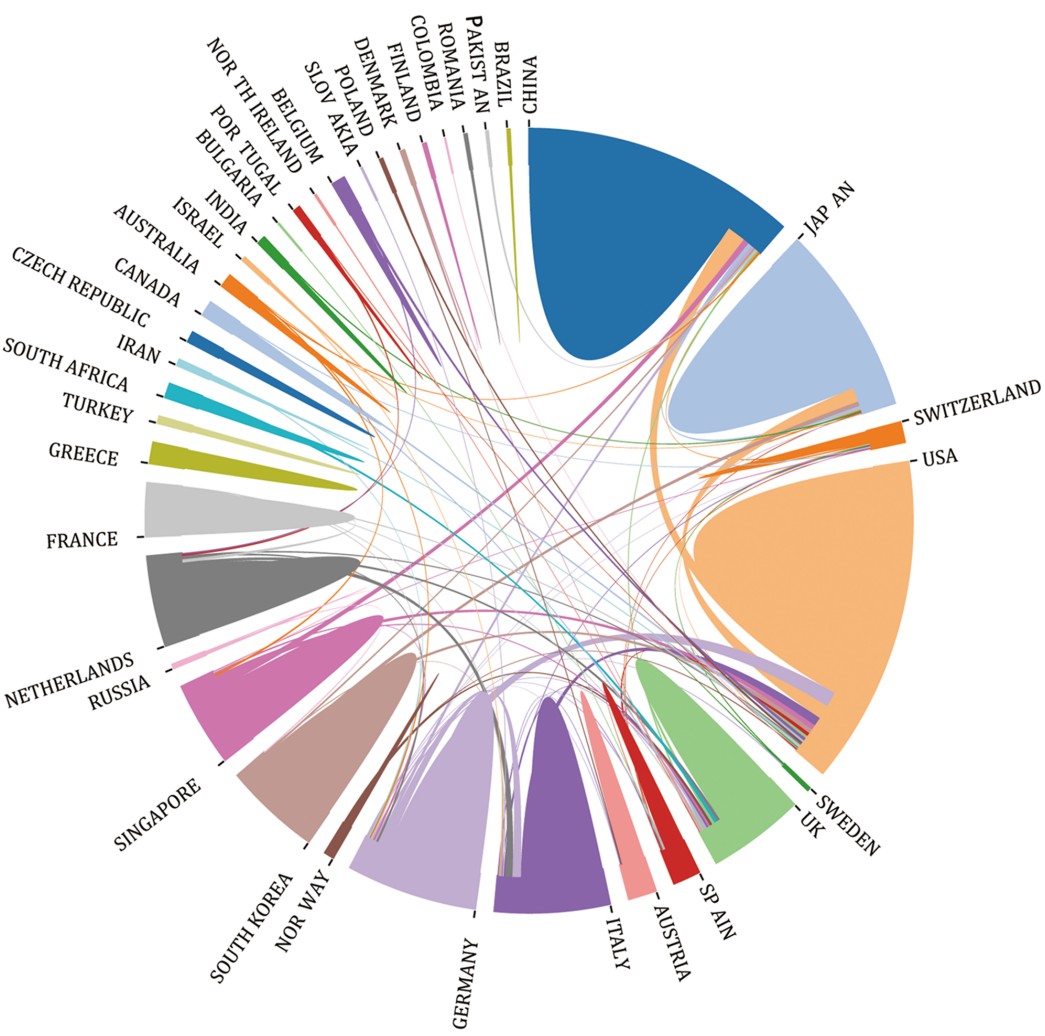

**Figure 3 Interactions between countries of the included articles.**

in the number of articles was identified for Japan. Among the international cooperation (Fig. 3), and the collaboration between the US and China was the most frequent, followed by US-Germany.

## Citation count and possible factors influencing citations

Among the 968 included articles, T100 publications on artificial livers were identified by WoS and ranked by the number of citations (Table 1; Table S1). The median number of citations was 198 (ranged from 47 to 394), with 20 papers cited over 100 times. The median number of citation index was 36.31 (ranged from 3.61 to 69), and there was strong correlation between citation index and the number of the citations ($r^2 = 0.343$, $p < 0.001$; Fig. S1A) per article in WoS. In addition, the number of citations and citation index of every article in Scopus database were strongly correlated, and the number of citation index between WoS and Scopus was also strongly correlated ($r^2 = 0.350$, $p < 0.001$; $r^2 = 0.990$, $p < 0.001$, respectively; Figs. S1B and S1C). Both the mean values and the

**Table 1  T100 most-cited articles ranked by the number of Times cited.**

| Rank | Articles | Publication year | Cited times (Web) | Citation index (Web) | Cited times (Scopus) | Citaitons index (Scopus) |
|---|---|---|---|---|---|---|
| 1 | Demetriou AA, Brown RS, Busuttil RW, et al. Prospective, randomized, multicenter, controlled trial of a bioartificial liver in treating acute liver failure. [J]. Annals of Surgery, 2016, 239 (5):667–670. | 2004 | 394 | 0.197 | 427 | 0.213 |
| 2 | Lee PJ, Hung PJ, Lee LP. An artificial liver sinusoid with a microfluidic endothelial-like barrier for primary hepatocyte culture [J]. Biotechnology & Bioengineering, 2007, 97(5):1340–1346. | 2007 | 220 | 0.110 | 236 | 0.118 |
| 3 | Kelm JM, Fussenegger M. Microscale tissue engineering using gravity-enforced cell assembly [J]. Trends in Biotechnology, 2004, 22(4):195–202. | 2004 | 191 | 0.095 | 200 | 0.100 |
| 4 | Medical applications of membranes: Drug delivery, artificial organs and tissue engineering | 2008 | 177 | 0.088 | 197 | 0.098 |
| 5 | Stamatialis DF, Papenburg BJ, Gironés M, et al. Medical applications of membranes: Drug delivery, artificial organs and tissue engineering [J]. Journal of Membrane Science, 2008, 308(1–2): 1–34. | 2006 | 161 | 0.080 | 162 | 0.081 |

Note:
See Table S1 for a complete list of T100.

standard deviations (SDs) of the number of citations and citation index for WoS and Scopus Database were evaluated. The mean values of the number of citations for these two databases were 79.270 (SD = 45.425,) and 83.000 (SD = 49.560), respectively, while the mean values of citation index were independently 0.040 (SD = 0.023) and 0.041 (SD = 0.025). The oldest cited paper was written by *Chan et al. (2004)* and was published in 2004. The latest paper written by *Larsen et al. (2016)* was published in 2016.

To identify the factors that determined the number of citations of T100 articles, we studied possible correlations between the number of citations and years since publication, authors, funding, and countries (Fig. 4). There were strong correlations of the number of citations with authors ($r^2 = 0.133$, $p < 0.001$) and countries ($r^2 = 0.275$, $p < 0.001$). While no correlations of the number of the citations with years since publication ($r^2 = 0.016$, $p = 0.216$), and funding ($r^2 < 0.001$, $p = 0.770$) were identified.

## Journals

More than 60 journals contributed to T100 publications, and T18 journals of T100 publications are listed in Table 2. Articles were most frequently published in Biomaterials ($n = 9$), followed by Tissue Engineering ($n = 7$), Liver Transplantation ($n = 6$), and Biotechnology and Bioengineering ($n = 6$). T18 journal IFs of T100 articles ranged from 1.59 to 17.016 with the median number of IFs (9.303). Many of the T100 articles were published in high-IF journals, however, these IFs were poorly correlated with the number of T100 articles ($r^2 = 0.023$, $p = 0.556$; Fig. S2A), and the number of average citations ($r^2 = 0,060$, $p = 0.345$; Fig. S2B).

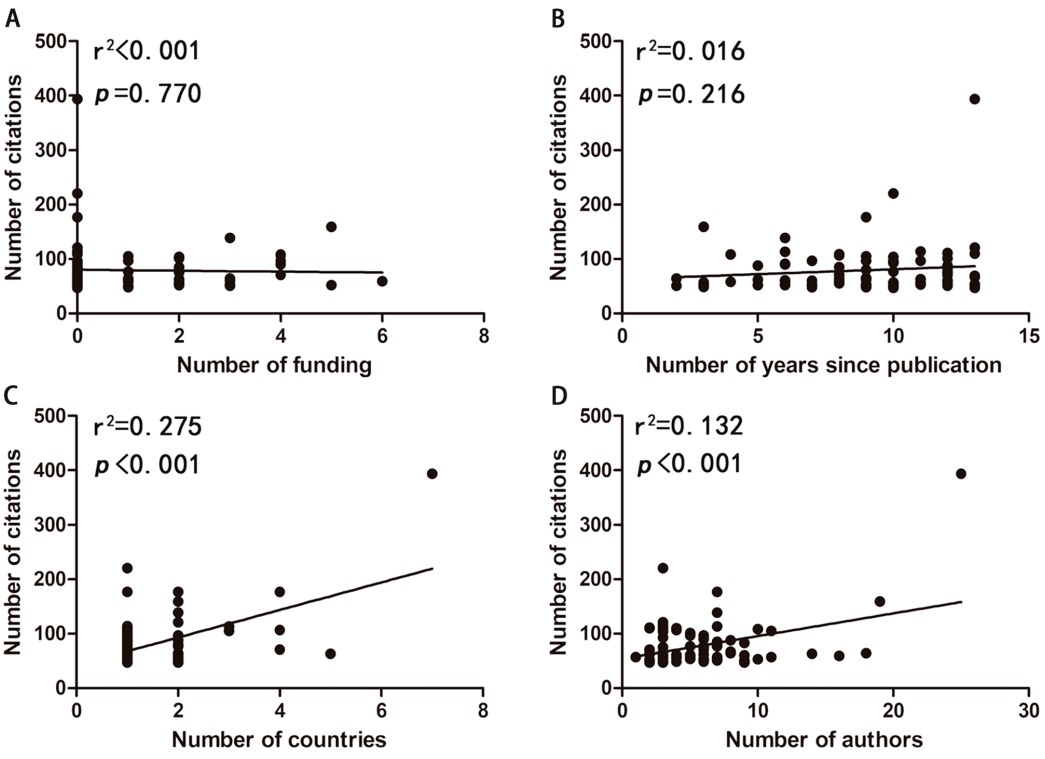

**Figure 4 Correlations between the number of citations and the number of funding (A), the number of years since publication (B), the number of countries (C), and the number of authors (D).**

## Authorship and institutions

More than 3,500 authors contributed to the included articles and the majority (89%) of T100 articles were produced by cooperation work involving ≥3 authors. The top five cited authors ranked by the number of articles were listed in Table 3. The most frequently appearing authors was Chamuleau, RAFM, who authored 32 included articles (two as first and five as corresponding author) with total 387 citations, followed by Li, LJ, who authored 31 included articles (three as first and 28 as corresponding author) with a total of 127 citations.

CiteSpace detected the information on author and co-cited authors and presented them through a network map (Figs. 5A and 5B). Among the cooperation between authors, Li, LJ ranked the first (cooperated with 30 authors), followed by Chamuleau, RAFM (cooperated with 28 authors). According to the network map of co-cited authors, Demetriou AA (210 citations) ranked first, followed by Nyberg SL (154 citations), and Van de K (153 citations).

Among the top five institutions of included articles (Table 4), the leading institutions with the most productive articles was University Amsterdam ($n = 85$, with 1,066 citations), followed by Zhejiang University ($n = 53$, with 160 citations). In addition, University Pittsburgh, National University Singapore, and Harvard University possessed 49, 44, and 40 included articles, respectively (with 211, 211, and 225 citations, respectively).

**Table 2  T18 Journals of the T100 publications were included.**

| Journal | No. of articles (citations) | Impact factors 2017–2018) |
|---|---|---|
| Biomaterials | 9(686) | 8.806 |
| Tissue engineering | 7(445) | 3.508 |
| Biotechnology and bioengineering | 6(536) | 3.952 |
| Liver transplantation | 6(502) | 3.752 |
| Journal of hepatology | 3(209) | 14.911 |
| Biomedical microdevices | 3(159) | 2.077 |
| Biotechnology letters | 3(138) | 1.846 |
| Annals of surgery | 2(504) | 9.203 |
| Journal of membrane science | 2(228) | 6.578 |
| Hepatology | 2(198) | 14.079 |
| Trends in biotechnology | 2(184) | 13.578 |
| Transplant immunology | 2(179) | 1.655 |
| Gut | 2(160) | 17.016 |
| Journal of cellular and molecular medicine | 2(157) | 4.302 |
| Journal of bioactive and compatible polymers | 2(154) | 1.598 |
| Tissue engineering part B-reviews | 2(121) | N/A |
| Stem cells | 2(115) | 5.587 |
| World journal of gastroenterology | 2(109) | 3.300 |

Note:
  N/A, not available.

**Table 3  The top five cited authors ranked by the number of articles.**

| Rank | Authors | No. of articles | First | Citations of first | Correspond | Citations of correspond | Total citations |
|---|---|---|---|---|---|---|---|
| 1 | Chamuleau, RAFM | 32 | 2 | 60 | 5 | 88 | 387 |
| 2 | Li, LJ | 31 | 3 | 26 | 28 | 122 | 127 |
| 3 | Hoekstra, R | 29 | 3 | 17 | 16 | 121 | 346 |
| 4 | Ding, YT | 24 | 0 | 0 | 19 | 97 | 113 |
| 5 | Ijima, H | 24 | 7 | 25 | 13 | 43 | 76 |

CiteSpace identified the information on institutions and showed them through a network map (Fig. 5C). Among the network map, Zhejiang University ranked the first with the cited numbers of 45, followed by Harvard University (with 33 citations), Kyushu University (with 33 citations), and University of Amsterdam (with 28 citations).

## Keywords and research fields

Keywords with bibliometric analysis provided information about directions and trend of research. A rough estimate of research changes could be found with Fig. 6. The keywords of "bioartificial liver" and "hepatocyte" appeared less and less during 2004–2017, however, "tissue engineering" was appeared more frequently after 2011, and almost became the most frequently used keywords during the period from 2015 to 2017.

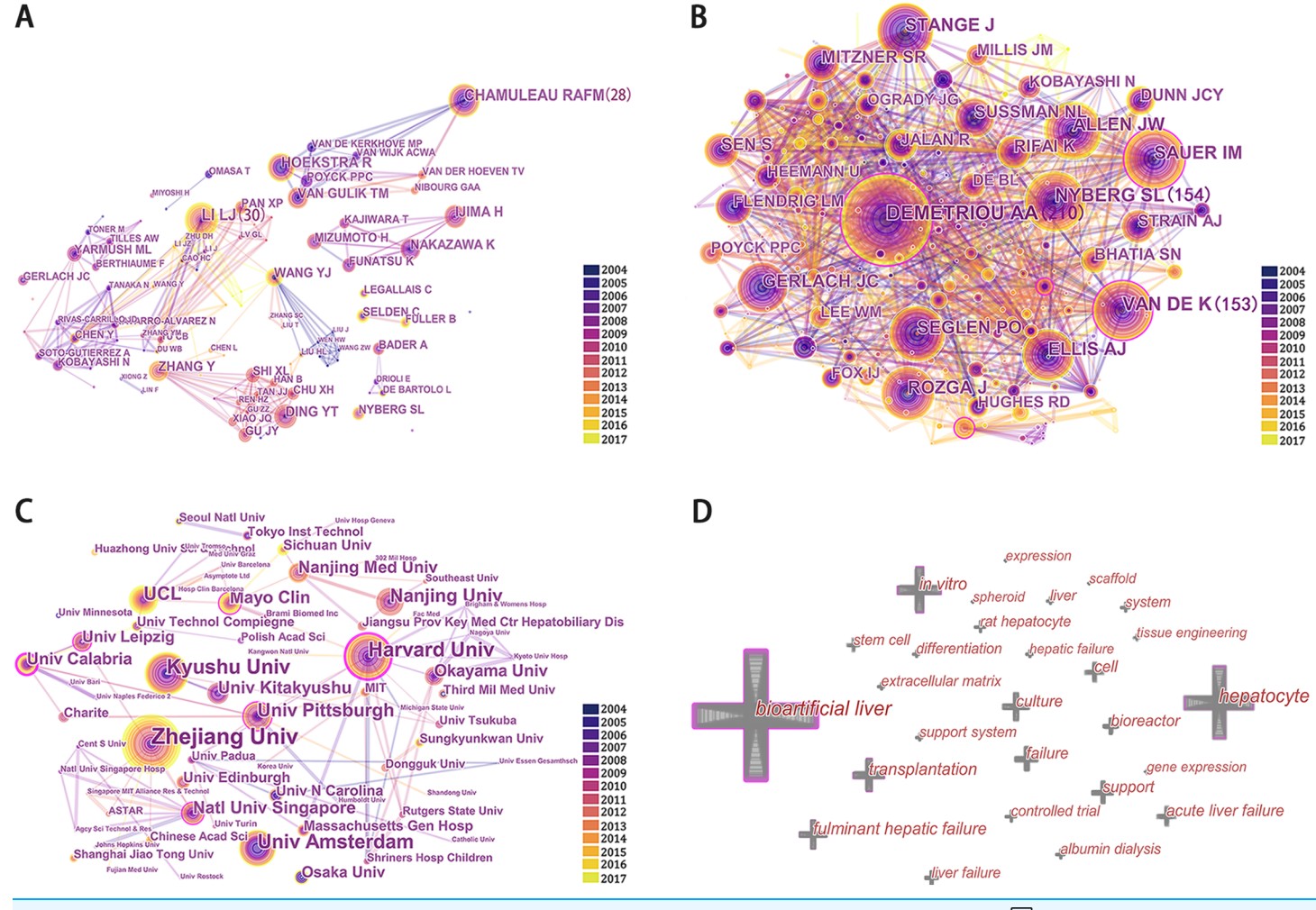

**Figure 5** Network map of authors (A), co-cited authors (B), institutions (C), and keywords (D).

**Table 4 The top five cited institutions of included articles on artificial livers.**

| Rank | Institution | No. of articles | No. of citations |
|---|---|---|---|
| 1 | University of Amsterdam, AZ Amsterdam, Netherlands | 85 | 1,066 |
| 2 | Zhejiang University, Hangzhou City, Zhejiang Province, Peoples of Republic China | 53 | 160 |
| 3 | University of Pittsburgh, Pennsylvania, US | 49 | 211 |
| 4 | National University of Singapore, Singapore | 44 | 211 |
| 5 | Harvard University, Cambridge, Massachusetts, US | 40 | 225 |

In addition, it was not difficult to find that each time the keyword of "acute liver failure" increase, either "bioartificial livers" or "tissue engineering" follows the rise.

Top 53 cited keyword and keyword plus (including some duplicates counts) ranked by the count (the total citations of the publications in which the keyword and keyword plus appeared) were identified and analyzed with CiteSpace (Fig. 5D; Fig. S1), among which

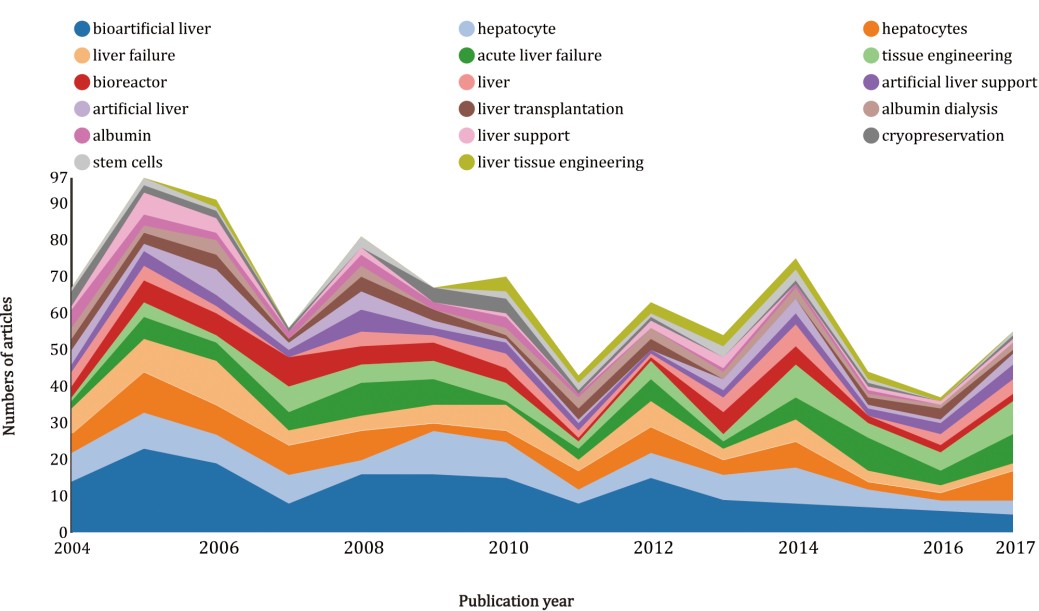

**Figure 6 Growth trends of keywords on artificial livers from 2004 to 2017.**

bioartificial liver ranked the first ($n = 405$), followed by hepatocyte ($n = 248$), in vitro ($n = 172$) and transplantation ($n = 151$).

Using CiteSpace software, we extracted top 50 cited (T50) keywords with the strongest citation bursts (Fig. S3), clearly indicating the research frontiers over time. The time interval was a blue line and the time period that represents a burst keyword category was a red line, suggesting the beginning and the end of the time interval of each burst (*Bornmann & Mutz, 2015*). The keywords with the strongest citation bursts were firstly ignited by "blood" from 2004 to 2007, followed by "blood purification" (2006–2010) and "hepatocyte like cell" (2008–2015). Keywords with the strongest citation bursts after 2010 were listed as follows: "prometheus" (2012–2013), "plasma exchange" (2014–2017), and "tissue engineering" (2014–2017).

More than 3,900 keywords were extracted from the 968 publications. After the data were combined with consent word, 1,732 keywords were ranked by the number of occurrences. Among them, 108 keywords appearing at least seven times, were identified and analyzed with Table 5. These keywords showed the main content of studies on artificial livers and research trend of this field. These keywords were classified into four domains based on two independent investigators (Meizhi He, Yan Li): bioartificial livers, blood purification, clinical, and other artificial organs. Among the 108 keywords, the bioartificial livers domain ranked the first with the highest percentage (57.40%). The blood purification domain was followed with 25.00%, the clinical was 14.81%, and other artificial organs was 2.78%.

The human liver technology topic of the human liver domain showed the highest percentage (15.74%), followed by human liver tissues and organs (14.81%). Among them, the most frequent keyword was "Amanitaphalloides" ($n = 95$) in hepatocyte related

**Table 5 Keyword on artificial livers from 2004 to 2017.**

| Domain | Topic | Percentage with keywords, % | Frequency of keyword occurrence (n) |
|---|---|---|---|
| Bioartificial livers domain (total, 57.40%) | Human liver technology | 15.74 | Academic medical center bioartificial liver (83), artificial liver (58), 3D cell culture (55), artificial liver support system (27), 3D models (26), Artificial (17), 3D co-culture (14), Hepatocyte cultivation (14), Liposome (12), engineered liver (11), artificial implantable devises (10), 3D (9), Genomics (8), Histology (8), 3D visualization (7), Cell culture (7), Human artificial mini chromosome (7), |
| | Human liver tissues and organs | 14.81 | Hepatocyte (39), adipose-derived stromal cell (18), Anatomy (14), Induced pluripotent stem (iPS) cells (12), Embryonic stem (ES) cell (11), Endothelial cell (10), Liver (9), Stem cell (9), adipose-derived stem cells (8), HepG2 (8), Liver progenitor cell (8), Mesenchymal stem cell (8), Bone marrow marrow mesenchymal stem cells (7), Feeder cells (7), Human hepatoblastoma cell line (7), High cell-density structures (7) |
| | Bioartificial livers device | 10.19 | Flat-bed configuration (28), Fluidized bed (24), Scaffold (10), Biomaterials (11), Bioartificial liver assist device (11), Hybrid artificial liver (10), Bioartificial organs (9), Bioartificial (8), Bioartificial liver (8), Xenotransplantation (8), PERV (8) |
| | Liver function index | 6.48 | A hepatic time (36), Autoimmune (25), Albumin synthesis (31), Growth factor (14), Glycosylation (10), Hepatic function (9), ALT (7) |
| | Hepatocyte related substance | 5.56 | Amanitaphalloides (95), Bilirubin (68), Alginate beads (62), Antioxidant (59), Endotoxin (29), Apheresis (20) |
| | Bioreactor | 4.63 | Airlift Reactor (78), Galactose (37), Flat membrane bioreactor (29), Bioreactor (12), Galactosylated membrane (9), |
| Blood purification Domain (total, 25.00%) | In vitro technique | 11.11 | Extracorporeal (30), Bioaffinity separation (18), Animal cell culture engineering (17), Hemodiafiltration (15), Fluid shear stress (14), Fractionated plasma separation and absorption (13), A combined rotational mold system (12), Molecular adsorb entre circulating system (12), Mass transfer (9), Extra corporeal liver perfusion (9), Hypothermia (8), Fluidization (7) |
| | Molecular substance | 7.41 | Ammonia (21), EROD (12), Medium (10), hemoglobin-based oxygen albumin (8), carrier (8), Acetaminophen (7), Midazolam (7), Glutathione S-transferase expression (7) |
| | Extracellular substance | 3.70 | Extracellular matrix (24), EGF (20), E-Cadherin (18), Hepatic growth factors (13) |
| | External device | 2.77 | Adsorption columns (17), Extracorporeal Liver Assist Device (14), Energy systems (9) |
| Clinical domain (total, 14.81%) | Clinical treatment | 8.33 | Major hepatectomy (16), intra operative shunt (14), Extended liver resection (13), Liver cell therapy (12), Liver support treatment (12), Gene therapy (10), Hepatocyte transplantation (8), Multi objective optimization (8), FDA guidelines (7) |
| | Disease | 6.48 | Acute liver failure (15), Alcoholic hepatitis (13), end-stage liver disease (11), acute-on-chronic liver failure (10), Hepatic failure (9), Hepatocellular carcinoma (8), Acute poisoning (7), |
| Other artificial organs domain (total, 2.78%) | Other artificial organs | 2.78 | Artificial heart blood pump (20), Artificial heart (clinical) (13), Artificial bone (12) |

substance topic, followed by "academic medical center bioartificial liver" (*n* = 83) in human liver technology topic, "airlift reactor" (*n* = 78) in Bioreactor topic, "bilirubin" (*n* = 68), "alginate beads" (*n* = 62), and "antioxidant" (*n* = 59) in Hepatocyte

**Table 6 The top five cited research area.**

| Rank | Research area | No. of articles |
|------|---------------|-----------------|
| 1 | Biochemistry | 30 |
| 2 | Applied microbiology | 24 |
| 3 | Cell biology | 21 |
| 4 | Gastroenterology & hepatology | 21 |
| 5 | Engineering | 16 |

related substance topic, "artificial liver" ($n = 58$) in human liver technology topic. Additionally, "extracorporeal" ($n = 30$), "hepatocyte" ($n = 39$), "a hepatic time" ($n = 36$), "albumin synthesis" ($n = 31$), "galactoses" ($n = 37$), and "3D cell culture" ($n = 55$) appeared 30 or more than 30 times.

While in an analysis of research area of T100 publications (Table 6), Biochemistry ranked first with the number of 30 publications, followed by Applied Microbiology ($n = 24$), Cell Biology ($n = 21$), and Gastroenterology & Hepatology ($n = 21$), respectively.

## DISCUSSION

### Total number of published items

This study is the first bibliometric analysis on artificial livers. The results of the number of articles each year had demonstrated that the research on artificial livers fluctuated from 2005 to 2014. There were twice descents in 2005–2007 and 2014–2016, probably for the following reasons: (1) the average annual incidence rate of liver failure decreased (*Ichai & Samuel, 2011*); (2) some investigators had finished their projects on artificial livers and changed their research orientation (*Sugawara, Nakayama & Mochida, 2012; Todo & Furukawa, 2004*); (3) WoS only covered a small part of publications (*Tanaka et al., 2011*). In fact, comparing with the growth trend of countries, the number of publications of each country remained stable, except for Japan which demonstrated a sharp decline since 2005. A possible explanation was that the incidence of diseases that can further develop into liver failure, such as acute liver injury, decrease by years in Japan and most researchers in Japan would like to investigate liver transplantation rather than artificial livers (*Ogura et al., 2010*). All in all, we concluded that the research on artificial livers was still progressing in a relatively slow pace but was extremely promising which requires researchers to solve the temporary problems in a better way.

### Country of origin and institutions

In growth trends of countries, the result showed that China played an important part in the research on artificial livers. This was probably due to the largest population of domestic hepatitis B patients (*Kjaergard et al., 2003*) and an undesirable liver transplantation situation in China (*Nielsen, 1998*). The high incidence rate of hepatitis B in China largely led to this increasing number of investigations of interest (*Chen, Shen & Xiang, 2011*). US, Japan, and Germany collectively contributed to a predominant number of publications,

verifying that countries with higher economic ranking were associated with larger quantity and better quality of biomedical publications (*Qiu et al., 2010*). In addition, USA, Japan, and Germany were developed countries and had a huge disease burden of liver failure, promoting more research interest and more research funding as well in artificial liver researches (*Gillum et al., 2011*). Figure 2B indicated that artificial live researches in the US were influential worldwide. The results of country of origin also lead to the results of institution. This finding confirmed that the great demand in effective treatment strategies could encourage the development of scientific researches in this field.

## Citation count and possible factors influencing citations

Among the 968 included publications, we extracted T100 publications and made a list in a table for advanced analysis. T100 publications on artificial livers were cited 47–394 times, with only 20 publications cited over 100 times. This number was quite small, for the reason that citations differed between different professional domains, mainly contributing to the number of researchers in specific medical fields (*Tang et al., 2016*). In *McDowell et al.'s (2017)* research on pediatric liver transplantation, the citations of T10 publications ranged from 175 to 635, significantly fewer than those of hypertension (2,242–7,248) (*Oh & Galis, 2014*) and diabetes (3,420–10,292) (*Zhao et al., 2016*). In order to minimize the effect of publication time on citations, the citation index was applied to estimate publications impact on their field in a short period. The results showed that there is high correlation between citations and citation index, indicating that citations were hardly affected by citation period. Additionally, high correlations between WoS and Scopus on citations were found with the premise that the citation index was strongly related to citations in Scopus. Besides, through investigating four possible factors that influence citations, the result demonstrated that the more authors and more countries work together, the higher quality of an articles could be. However, this phenomenon contradicted *Ahmed et al. (2016)* research, which concluded that two-authored article was the most suitable. It seemed that the effect of the number of authors differs between specific research fields. No correlation between the number of citations and the number of years since publication was identified, which might be related to the tendency of citing particular papers in researchers (*Azer & Azer, 2016*). There was also no correlation found between the number of citations and the number of funding, indicating that the scientific impact of researchers was only weakly limited by the number of funding (*Fortin & Currie, 2013*).

## Journals

Numerous studies had shown that the impact factor of journal was the best indicator for citations (*Saha, Saint & Christakis, 2003*). However, citations were not clearly affected by IFs of the journals in this analysis (*Hecht, Hecht & Sandberg, 1998*). They found that the majority of the considered as "highest-impact journals" did not report fresh research results. The issue that caused the most trouble was that IF was a quantitative measure of a quality that cannot be quantified.

## Authorship

Analysis of authorship describes the cooperation between authors and the high productive authors, of which the top five cited authors listed, produced more than 400 publications. Thus, they were called "productive authors" (*Zongyi, Dongying & Baifeng, 2016*). Unfortunately, a small number of prolific authors appeared in the network map of co-cited authors, suggesting that not only the number of publications but also the quality of publications should be assessed in the evaluation of these authors. Demetriou AA, Nyberg SL, and Van de K were cited more than 100 times. Although none of them belong to the prolific authors, they played an important part in artificial liver research.

## Keywords and research fields

Keyword analysis consists of three parts: (1) a rough estimation of the trend of research directions (2) analysis of the research hotspots and frontiers (3) detailed classification and analysis of research area. The increase of "bioartificial livers" and "tissue engineering" usually followed by the increase of "acute liver failure," because acute liver failure usually developed into serious liver failure, and artificial livers were considered to be the most effective treatment. The keyword of "bioartificial livers" with more than 400 citations decreased during the period from 2004 to 2017. It reached a peak (54 articles) at 2005, and most of them focused on the topic of hollow fiber (*Abu-Absi et al., 2005*; *Lu et al., 2005*; *Nguyen, Brotherton & Chau, 2005*), bioartificial cell (*Aoki et al., 2005*; *Cheng et al., 2005*; *Gerlach, 2005*; *Monga et al., 2005*), and transplantation (*Garkavenko et al., 2005*; *Liu & Chang, 2005*). This finding suggested that many researchers showed great interest in bioartificial livers in 2005. They made every effort to culture human hepatocytes, rat hepatocytes, and stems cells to support liver functions and improve the device of bioreactors (*Park et al., 2005*). The keyword "bioartificial livers" covered the meaning of "hepatocyte," and thus the trend of "hepatocyte" was similar with "bioartificial livers." Tissue engineering studies on 3D technology (*Arai et al., 2017*), mathematical model (*Chapman et al., 2017*), liver microencapsulation technique (*Chapman et al., 2017*) and stems cells (*Kadota et al., 2014*) were mainly carried out in 2016 and 2017. According to T50 keywords of artificial livers, we extracted the top three research hotspots and listed them as follows:

  i) Bioartificial liver: A bioartificial liver system incorporated hepatocytes into a mechanical, albumin dialysis (AD)-based artificial liver support device to replace liver function (*Nicolas et al., 2017*). The two main elements of bioartificial livers were bioreactors and cell material. There was no definition of which kind of cell is the most ideal, and many researchers were still in search of the most ideal cells.

 ii) Hepatocyte: Hepatocyte was known as the function unit of liver. It was ideal to investigate a kind of hepatocyte with regeneration ability and necessary functions for substitution of necrotic or dysfunctional hepatocytes.

iii) Transplantation: With the development of liver regenerative medicine, cell transplantation was becoming an increasingly popular topic, and many researchers showed great interest in it.

The keywords bursts were considered as research frontiers over time. We extracted some keyword burst and divided them into three parts as follows:

i) Blood and blood purification: Acute liver failure would be found variety of toxic substances in blood. Therefore, researches on artificial livers mainly focus on extracorporeal blood purification (hemodialysis, hemofiltration, hemodiafiltration, plasmapheresis, hemodsorption, cell-based therapy, etc.) (*Thongboonkerd, 2010*; *Nie et al., 2015*).

ii) Plasma exchange and Prometheus: Since relatively simple detoxification devices use in 1999 (AD, MARS) and 2003 (fractionated plasma separation, Prometheus), the treatment of liver failure had improved a lot (*Davenport et al., 2015*). However, prospective trials of extracorporeal support with AD, superflux dialyzers in series with absorption columns, and bioartificial devices containing hepatocytes have not demonstrated a significant survival advantage. Therefore, researchers found that before liver failure, plasma exchange improved cardiovascular stability, extending survival, and increase overall survival.

iii) Tissue engineering: "Tissue engineering" appeared in 2014 for the first time and continued until 2017. It is the best prediction that liver function could be restored with the technology of tissue engineering, and even now, many researcher are working hard in perfecting the technology of tissue engineering. The future is bright for patients with serious liver failure in receiving a "new" liver through tissue engineering.

Among the four domains mentioned, the domain of "bioartificial livers" occupied the largest portion and the other artificial organ domains occupied the smallest portion. This suggested that bioartificial liver would still be a treatment strategy of great potential in liver failure and needed more researchers to promote its development. At the same time, the occupation of clinical domain suggested that clinical requirement was one of the strongest motivations for stimulating the development of artificial livers. The result of the research fields was similar to the four domains of keywords.

There were some limitations in this study. Firstly, the data were only extracted from the WoS database. Secondly, it was difficult to make definite predictions on the development trend as well as the most popular topic of interest since researches on artificial liver was still developing slowly though with great potential. Despite its limitations, this study has provided the tendency and main context of researches in artificial livers, indicating that hepatocyte transplantation might be the frontier in artificial livers but now the liver support system could gain better clinical effect.

## CONCLUSIONS

The number of publications on artificial livers fluctuated from 2004 to 2017, most of which are Chinese publications. After excluding several confounding factors (database, number of authors, countries, years since publication, and funding), the citations roughly estimated the quality of the included articles. The analysis of authorship and institutions

also contributed to evaluating the quality of the included articles. In addition, through a detailed analysis which roughly assessed the research tendency, hotspots and frontiers of keywords, this study showed the progress and research trend of artificial livers. However, although it seems that the future of artificial livers seems brighter for hepatocyte transplantation, the systems of artificial livers now are more focusing on blood purification, plasma exchange, etc.

## ABBREVIATIONS

| | |
|---|---|
| **WoS** | Web of Science |
| **T18** | The top 18 cited |
| **Ifs** | Impact Factors |
| **T100** | The top 100 cited |
| **T5** | The top 5 cited |
| **T53** | The top 53 cited |
| **T50** | The top 50 cited |
| **JCR** | Journal Citation Reports |
| **MARS** | Molecular Adsorbent Recirculating System |
| **AD** | albumin dialysis |
| **FPS** | fractionated plasma separation |
| **SD** | standard deviation |

## ACKNOWLEDGEMENTS

The authors would like to thank editors and the anonymous reviewers for their valuable comments and suggestions to improve the quality of the paper.

### Funding

This work was supported in part by grants from the Undergraduate Innovation and Entrepreneurship Training Program (201712121157), the China Postdoctoral Science Foundation (CPSF, No. 2018T110855 and No. 2017M622650) and the National Natural Science Foundation of China (NSFC, No. 81300370). There was no additional external funding received for this study. The funders had no role in study design, data collection and analysis, decision to publish, or preparation of the manuscript.

### Grant Disclosures

The following grant information was disclosed by the authors:
Undergraduate Innovation and Entrepreneurship Training Program: 201712121157.
China Postdoctoral Science Foundation (CPSF): No. 2018T110855 and No. 2017M622650.
National Natural Science Foundation of China (NSFC): No. 81300370.

### Competing Interests

The authors declare that they have no competing interests.

## Author Contributions

- Yan Li conceived and designed the experiments, performed the experiments, analyzed the data, contributed reagents/materials/analysis tools, authored or reviewed drafts of the paper, approved the final draft.
- Meizhi He conceived and designed the experiments, performed the experiments, analyzed the data, contributed reagents/materials/analysis tools, authored or reviewed drafts of the paper.
- Ziyuan Zou performed the experiments, analyzed the data, authored or reviewed drafts of the paper.
- Xiaohui Bian performed the experiments, analyzed the data, authored or reviewed drafts of the paper.
- Xiaowen Huang prepared figures and/or tables, authored or reviewed drafts of the paper.
- Chen Yang prepared figures and/or tables, authored or reviewed drafts of the paper.
- Shuyi Wei prepared figures and/or tables.
- Shixue Dai conceived and designed the experiments, contributed reagents/materials/analysis tools, approved the final draft.

## Data Availability

  Raw data is provided in the Supplemental Files.

## Supplemental Information

Supplemental information for this article can be found online at http://dx.doi.org/10.7717/peerj.6178#supplemental-information.

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
