# Peer review of "Artificial liver research output and citations from 2004 to 2017: a bibliometric analysis"

_PeerJ, doi:10.7717/peerj.6178_

## Round 0.1 · original submission · Minor Revisions

Please address the reviewers' comments in detail.

Reviewer 1 ·

Basic reporting

no comment

Experimental design

no comment

Validity of the findings

no comment

Additional comments

This paper performed the bibliometric analysis on publications in artificial liver research. It quantitatively investigated the research progression of an important area, and reported its research trend from different perspectives. The paper also showed the high correlation between citations with author and countries. The paper is well organized and well written, but I still have a few of concerns and listed below.
Major:
1. I am confusing about the searching criteria. As stated in introduction, the researchers of “artificial livers” are categorized into three groups, one of which is “bioartificial liver”. In the paper searching, both “artificial liver” and “bioartificial liver” were used. I am wondering why the other two terms “blood purification” and “hepatocyte transplantation” were not included. Or I would like to suggest using only “artificial liver” and/or its related words (artificial livers et al).
2. Also in “search strategy”, which fields the searching were performed on? Is the searching performed on one field (like keywords, abstract, title), or multiple fields?
3. In “inclusion and exclusion”, criteria (2) is “any type of researches”. I would like to suggest the authors to explain a little bit about the “type” of researches, is it about journal/conference, or abstract/regular paper, or anything else?
4. I would suggest the authors to explain the “citation index” as it is not straightforward for other readers to get what it is to measure.
5. In “journals and publication year”, Fig S2 A-B presented the correlation between IF with number of article and number of citation. I have several questions about this subsection “journals and publication year” and this figure listed as below. 1) I didn’t find the information about “publication year” in this paragraph. 2). I am not clear about what information did figure S2 present. There are both 17 points in A and B, each point represent one journal if my understanding is correct? If so, what do the x-axis represent? It is the total statistics for each journal? I don’t think Fig S2 is readable and I also don’t think it is reasonable to get the conclusion that the top 100 publications were published in low IF journals from these two figures, if the number of citation of each journal is calculated by summing the citations of each paper from this journal. Because if using the sum of citations from each publication, the number of papers will affect the correlation. Instead, I would suggest to use the average measurement.
6. I would like to suggest the authors to add details to the “statistical analysis”, like introducing some terms (e.g., citation index), and present the statistical analysis more precisely (for example, it is not clear to say “correlation between citation and authors”, it would be better if say “correlation between the number of citations and the number of authors in each publication”; and to include more details in figure and table captions, making them be more readable.
7. In discussion, the author mentioned the possible reasons of decreasing trend of artificial liver. Can the authors help provide literature or evidence for supporting the reasons listed?
Minor:
1. Line 221: there is an error sentence.

·

Basic reporting

• English language: ‎

Actually, I didn’t face any problem in understanding the content of the manuscript. The ‎English language was clear, professional and understandable. It was presented in all parts of ‎the article in a distinct and unambiguous manner. ‎

• The structure of the manuscript, figures, and tables comply with the standards reported by Peerj. It seems to me that the figures and tables are well designed and labeled and correlate with the content of the manuscript and ‎summarize the key findings and trends in a way that is easily interpreted and comprehended by ‎the reader without returning to the main text. Captions, table headings, and plotted parameters in ‎graphs are described appropriately enabling the reader to understand the data easily. ‎

• Reference section:
Authors may consider enumerating the full list of authors with their initials in all references based on the format suggested by PeerJ.

• Title: ‎
The title of this article precisely portrays the main essence of the study. The reader can easily ‎understand what the manuscript is about.‎

• Abstract:‎
The abstract is an essential part of a research paper because it gives the reader a brief review of ‎the study. Herein, the abstract is well written summarizing the main points and findings ‎reported in the main text. It includes all the keywords and the needed information that is ‎easily perceived by researchers outside the specialty of liver transplantation. ‎

• Keywords: ‎
I think the keywords mentioned in the paper represent the content of the manuscript and ‎facilitate the retrieval of the article. ‎

• Introduction section:‎

‎>>>Positive points: ‎
The introduction formulates the research question that will be addressed in the manuscript. It ‎gives the reader a good definition of bibliometric science, explaining its influence in medical ‎literature and benefits for future researchers. It also clarifies what is citation analysis, which is one ‎of the most important methods in bibliometric analysis, and states how articles are assessed ‎based on this measure. Study aims and objectives are well identified in the last paragraph of the ‎introduction. Furthermore, the authors clearly state the importance and the reasons behind ‎executing this study. ‎

‎>>>Recommendations/suggestions:‎
‎ Obviously, this research article is generally about artificial liver devices. It is apparent that you ‎mention the history and classifications of the artificial liver system. However, to enhance the quality of ‎the introduction, I recommend enclosing sufficient background information about the K. Onodera ‎et al.’s definition of artificial liver used in this analysis, the purpose of this approach and how it ‎works. ‎

Experimental design

The research article conforms to the aims and scope of PeerJ. It is a medical research review study that statistically analyzes the citations and research output for artificial livers publications between 2004 and 2017.

Research questions, aims, and goals are distinctly identified in the introduction section. They are consistent with the main text. Authors explain how this analysis will help future researchers in their work.

• Methods section: ‎

‎>>>Positive points: ‎
It is distinctly enumerated the research platform used in performing this study. Authors specify the ‎range of dates and all potential keywords utilized in conducting the analysis. Also, the date of final ‎data retrieval is also written in this part. ‎
Inclusion and exclusion criteria are well listed in the manuscript. Two reviewers with their names ‎were chosen to screen the studies based on titles, abstracts, and full-text versions. Also, a third ‎independent investigator participated in solving any discordances between the two reviewers. ‎
Evaluation of the included studies necessitates an extraction of multiple parameters that are ‎plainly mentioned in this section.‎
Authors depicted the statistical tools that are applied to carry out the statistical and bibliometric ‎analysis.

‎>>>Recommendations/suggestions:‎
‎1)‎ My suggestion is to demonstrate which investigators engaged in retrieving the relevant ‎data from the included studies. ‎
‎2)‎ References for the statistical and bibliometric methods should be added to the main text. ‎

Validity of the findings

• Results section: ‎

‎>>>Positive points: ‎
Prisma diagram is clearly illustrated showing how many articles retrieved from the original search, ‎how many articles screened, the number of articles included, and the number of articles excluded ‎with documented reasons. ‎
The results are well explained and presented in a proper format. The data are robust, statistically significant, and answer the ‎main research question.‎ P value and R2 are calculated when comparing ‎the correlation between the median value of the number of citations and the median value of citation ‎index as well as the correlations between the number of citations and funding, years since ‎publication, countries, and authors.

‎>>>Recommendations/suggestions:‎
‎1)‎ It would be nice if you calculate the mean values and standard deviations of the number of ‎citations and citation index for both WoS and Scopus Database. ‎

‎2)‎ There are some discrepancies between the results in the main text and the results in the ‎tables and figures. For instance;‎
a)‎ In lines 227-228, you say that the Human liver tissues and organs topic of human liver ‎domain showed the highest percentage (14.81%), whereas, in Table 5, it is indicated that ‎Human liver technology has the highest percentage (15.74%).‎

b)‎ In lines 230-231, Airlift reactor (n=78) in Bioreactor topic should be reported after ‎‎‘academic medical center bioartificial liver and before ‘bilirubin’.‎

c)‎ In lines 233-234, Galactose (n=37) in bioreactor should be added to the keywords that ‎appeared more than thirty times

d)‎ In line 171, it is denoted that the range of impact factor of T18 journal of T100 articles ‎is 1.31 to 16.658. While in table 2, the maximum factor is 17.016 for GUT. ‎

• Discussion section:‎
‎>>>Positive points:‎
Generally, the discussion suits with the aims proposed in the introduction section. Authors ‎conversed about the findings and provide potential explanations and interpretations for data. ‎Also, some results in this analysis are compared to the results of other studies. ‎
Limitations of the study are clearly reported at the end of the discussion part. The authors’ ‎conclusions are properly stated and supported by their evidence and findings.

‎>>>Recommendations/suggestions:‎
‎1)‎ It seems to me that there are two sharp drops in the total number of publications between ‎‎2005 and 2017. I think authors need to touch the second drop (at 2016) and provide ‎possible explanations for it as they did for the first drop (at 2007).‎
‎2)‎ It would be great if authors could state the reasons behind the substantial contributory role ‎of USA, Japan, and Germany to artificial liver research as they did for China.‎
‎3)‎ The results of Zhejiang University in line 262-263 are already mentioned in the results ‎section and I don’t think it is necessary to repeat them in the discussion part. ‎
‎4)‎ The sentence about Hepatitis B incidence in China in line 263-264 is duplicate and this ‎information could be added to line 256-257.‎
‎5)‎ In line 260-261, the worldwide influence of USA research is indicated in Figure 4 and not ‎in Figure 2B.‎
‎6)‎ ‎ In line 281, it is pointed out that four factors that impact the citation number have been ‎investigated. However, only three factors (number of authors, number of countries and ‎the number of years since publication) have been addressed. My recommendation is to ‎explain why the fourth factor (number of funding) does not affect the number of citations in ‎this analysis. ‎
‎7)‎ The findings of Ahmad et al in line 284 need to be noted and explained how they differ ‎from the authors’ premise. ‎
‎8)‎ The impact factor is considered a good indicator for assessing the quality of studies. ‎Authors may consider giving more details about the ineffectualness of impact factor in this ‎analysis. ‎
‎9)‎ Authors may need to describe adequately how their findings will affect the future research. ‎

---

## Round 0.2 · accepted · Accept

All the reviewers' comments have been properly addressed.

Reviewer 1 ·

Basic reporting

no comment

Experimental design

no comment

Validity of the findings

no comment

Additional comments

The authors have responded to all my questions clearly. I have no further comments/suggestions.

·

Basic reporting

No comments. All suggested changes have been performed well.

Experimental design

No comments. All suggested changes have been performed well.

Validity of the findings

No comments. All suggested changes have been performed well.